# Predictive MRI Biomarkers in MS—A Critical Review

**DOI:** 10.3390/medicina58030377

**Published:** 2022-03-03

**Authors:** Vlad Eugen Tiu, Iulian Enache, Cristina Aura Panea, Cristina Tiu, Bogdan Ovidiu Popescu

**Affiliations:** 1Neurology Department, “Carol Davila” University of Medicine and Pharmacy, 050474 Bucharest, Romania; cristina.panea@umfcd.ro (C.A.P.); cristina.tiu@umfcd.ro (C.T.); bogdan.popescu@umfcd.ro (B.O.P.); 2Neurology Department, Elias University Emergency Hospital, 011461 Bucharest, Romania; 3Neurology Department, University Emergency Hospital of Bucharest, 05009 Bucharest, Romania; ion-iulian.enache@rez.umfcd.ro; 4Neurology Department, Colentina Clinical Hospital, 020125 Bucharest, Romania

**Keywords:** MRI, biomarkers, multiple sclerosis, prediction

## Abstract

*Background and Objectives:* In this critical review, we explore the potential use of MRI measurements as prognostic biomarkers in multiple sclerosis (MS) patients, for both conventional measurements and more novel techniques such as magnetization transfer, diffusion tensor, and proton spectroscopy MRI. *Materials and Methods:* All authors individually and comprehensively reviewed each of the aspects listed below in PubMed, Medline, and Google Scholar. *Results:* There are numerous MRI metrics that have been proven by clinical studies to hold important prognostic value for MS patients, most of which can be readily obtained from standard 1.5T MRI scans. *Conclusions:* While some of these parameters have passed the test of time and seem to be associated with a reliable predictive power, some are still better interpreted with caution. We hope this will serve as a reminder of how vast a resource we have on our hands in this versatile tool—it is up to us to make use of it.

## 1. Introduction

Multiple sclerosis (MS) is an inflammatory neurodegenerative disease of the central nervous system that poses a challenge to clinicians due to its remarkable inter- and intra-individual heterogeneity [1].

MS still lacks specific humoral biomarkers for diagnosis, prognosis, or progression, but data derived from magnetic resonance imaging (MRI) measurements might represent our best predictive biomarkers to date.

Indeed, while the role of MRI in the diagnostics of MS is unquestionable, we are still exploring what other data might be derived from this investigation. Almost all aspects of correctly managing MS patients rely on corelating clinical evolution with MRI scans, from initial disease modifying drug (DMT) choice to assessing DMT efficacy, identifying subclinical activity of disease or progression, and many more.

Much research has been aimed at finding predictive imagistic biomarkers for MS, with both white matter and gray matter metrics having been evaluated in numerous studies for their capacity to predict disease evolution.

This review will focus almost exclusively on the prognostic value associated with each of the major MRI measurements in use today.

## 2. Materials and Methods

For the purpose of this review, studies regarding MRI predictive biomarkers for all disease phenotypes (relapsing–remitting MS—RRMS; progressive MS—PMS) were considered eligible.

A literature search was performed in PubMed, Medline, and Google Scholar, from inception to 31 December 2021. The main search terms included were “multiple sclerosis”, “MRI”, “predictive”, and “biomarkers”. Papers written in languages other than English were excluded. References from the selected articles were then screened for further records. The authors independently assessed the selected articles to evaluate their eligibility, and disagreements were solved by discussion.

## 3. Results

### 3.1. The Need for Prognosis Biomarkers in MS

For years now, there has been an ongoing debate whether escalation or induction therapy is better for MS patients. Escalation therapy is defined as starting with a low to moderate efficacy DMT and escalating, if needed due to poor control of the disease, to a second-line therapy. Induction therapy differs from the former by starting out patients with high-efficacy DMTs and switching only at a later time to a maintenance, first-line agent.

Recent data seem to suggest induction therapy should be favored [2,3,4,5]. Notably, The European Committee for Treatment and Research in Multiple Sclerosis (ECTRIMS) recently published a review on aggressive MS treatment [6] and a guideline on treatment choice in MS encouraging more aggressive DMT choices at first (unpublished at the moment of writing this article, presented as “Update of the ECTRIMS/EAN Guidelines on the Treatment of Multiple Sclerosis. Updated recommendations” by the Steering Committee for the update and upgrade of the ECTRIMS/EAN guideline on the pharmacological treatment of people with Multiple Sclerosis, 15 October 2021, during the 37th ECTRIMS Congress).

How quickly this will be implemented is a different matter. Patients’ choice, intolerable side effects, pregnancy therapy restrictions, and the added burden to healthcare budgets are just some of the obstacles that stand in the way [7].

For now, most MS centers will choose a hybrid strategy—patients who present highly active or aggressive forms of MS will be started on second-line treatment options, while those who have inactive, low-risk forms will be started on more modestly effective therapies such as interferons or glatiramer acetate.

Unfortunately, some cases will be challenging to classify regarding disease activity and predicting progression following diagnosis remains a difficult task.

Can this decisional process rely on the biomarkers we have available today?

### 3.2. How Do We Define Prognosis and What Is a Bad One?

When talking about MS, a clinician will judge the prognosis of each case based on disease activity and risk factors for a poor evolution (more on that later), while also considering the current burden of the disease, clinical subtype, and many other factors.

We usually communicate this prognosis to our patients as the amount of time it will take for them to reach a certain amount of disability, usually the moment they will require a walking aid such as a cane or crutch.

Assessing this risk, however, is quite difficult, as each factor involved is under some degree of uncertainty or controversy. For example, there is no universal definition for “aggressive” MS.

In 2018, an ECTRIMS Focused Workshop on Aggressive MS tried and failed to define this term due to lack of available data correlating severe disease with imaging and molecular biomarkers [8].

Whether it is called “aggressive”, “highly active”, or “malignant” [9,10,11], most definitions (spanning decades) usually agree that it should be a rapid deterioration to a certain EDSS (usually to a score of 6.0 over 5 to 10 years), with some authors considering other conditions such as the number or features of relapses (aggressiveness, sequelae, EDSS impact, certain functional systems involved, etc.) or failure of DMTs. Others considered that a time of 3 years from RRMS onset to SPMS phase would also qualify as aggressive. Most authors will also include MRI features in defining aggressive MS (with gadolinium-enhancing lesions and new T2 lesions being key markers most of the time) [12,13,14,15,16,17,18].

The other end of this spectrum is the holy grail of managing MS—the “NEDA-4” status. Standing for “No Evidence of Disease Activity”, NEDA-4 is a concept that evolved over time by adding more items to the previous definitions (there was a NEDA, a NEDA-3, and now this). It is currently defined as no evidence of relapses, new or enlarged T2 lesions, and 6-month confirmed disability progression (defined as an increase in EDSS score of 1.5 points from a baseline score of 0, of 1.0 point from a baseline score of 1.0 or more, or of 0.5 points from a baseline score of greater than 5.0). The mean annualized rate of brain volume loss should also be less than 0.4% [19,20,21,22,23].

Some authors contest the validity of the NEDA-4 concept due to the tools we have on our hands to define it with, and they may have a point.

The very concept of defining and measuring disability and disability progression in MS is still flawed to some degree, and even the most used scales today—the EDSS (Expanded Disability Status Scale) and the MSFC (Multiple Sclerosis Functional Composite)—have important limitations.

The EDSS falls short on some important aspects, such as its non-linear progression, bimodal population distribution (distribution grouping around the scores of 3 and 6), irregular progression between intervals, measuring different aspects of disability at different points along the scale, inter- and intra-rater reliability issues, and poor to moderate correlation with MRI measures [24,25,26,27,28]. The EDSS has also received criticism for being imprecise at the lower end of the scale, insensitive at the middle and upper ends, and too heavily dependent on ambulation; not to mention that the upper extremity and cognitive functions are insufficiently assessed, that the cerebellar functional system has a very limited contribution to the score, and the list goes on [29,30].

The MSFC, which was specifically developed to overcome these problems, also has issues with a noticeable learning effect, poor patient acceptance (especially for the PASSAT testing), not being recognized by regulatory agencies as a primary disability outcome measure, and also lacking visual testing and still falling short in correlating with other MS measures such as the MRI [27,31].

It is because clinicians, patients, and studies alike define prognosis in MS today by the time elapsed to reach a certain degree of disability or to reaching a continuous progression of disability for a sustained period of time that it is of paramount importance to correctly define and track disability in MS. We must be mindful of these important shortcomings, as many studies have investigated the correlation of clinical measurements with different imagistic and humoral biomarkers.

Another big issue with our current stance on aggressive MS is that all our definitions are retrospective. Waiting for lesion and disability accrual before taking action is not a good strategy and that is why MS is in dire need of prognosis biomarkers.

### 3.3. Are Prognosis Biomarkers in MS Even Possible?

Prognosis biomarkers in MS were a rather controversial term, since MS tends to be quite an unpredictable disease [32]. In the long period of time that has elapsed since MS was first described, many tried to find risk factors for a poor prognosis. The fact that almost all these attempts have now been long forgotten is testimony to the difficult task ahead.

Kurtzke made an attempt at this with the “five-year rule”, stating that patients who had minimal accumulated disability following the first five years of disease evolution faced more favorable outcomes—needless to say, this has since been disproven [33].

Are prognosis biomarkers in MS even possible? The answer is probably “yes”, and we have had one of those biomarkers available for decades now.

### 3.4. The Use of MRI Metrics as Prognosis Biomarkers

The advent of MRI scans in the 1980s brought a revolution to the world of MS [34], with MRI criteria quickly being developed and standardizing the diagnostic process [35]. Research into how MRI data can be used in MS is still driving forward our understanding of the disease today.

Many MRI parameters have been correlated with MS, arguably the most popular of which are white matter lesions (contrast enhancing lesions, new lesions on longitudinal scans, and total white matter lesion volume and number) and cerebral and spinal volumetrics, with gray matter disease being a hot topic in recent years.

### 3.5. Evaluating White Matter (WM) Pathology in MS

WM lesions have been used as biomarkers for the prognosis and progression of the disease for a long time, with WM lesions often being one of the most important factors in guiding DMT choice. For short-to-medium term, baseline MRI scans have been considered by most clinicians to give the most accurate predictions of all biomarkers and have been used in guiding DMT choice [36,37,38,39,40,41].

Classic MRI measures that evaluate WMLs in MS, using conventional techniques (T1, T2, fluid-attenuated inversion recovery (FLAIR), etc.), usually refer to the number and volume of gadolinium-enhancing (GdE) lesions, as well as hyperintense lesions on T2-weighted scans and hypointense “black holes” on T1-weighted scans [42].

MRI scanners have been getting better and better at detecting WMLs [43,44]. Limitations still exist, however, as some authors have shown that T2/FLAIR WMHs overestimate neuropathologically confirmed demyelination in the periventricular areas but underestimate it in the deep WM [45], and overall sensibility and specificity hovers around 80% to 90%. As with all MRI measurements, higher field strengths and resolutions (3D versus 2D) will produce better results [46].

A significant number of lesions visible on MRI go undetected clinically. Studies have shown that, even when assessing conventional sequences at 1.5T MRI scans, subclinical pathological processes might be 5 to 10 times more active than clinically expected [47].

#### 3.5.1. T1 Black Holes

Some T2 lesions appear dark on T1-weighted spin-echo images—known as “black holes”. They are usually classified as either acute black holes, which tend to diminish or vanish in about 6 months, or persistent black holes, which persist for a long time (as they represent irreversible axonal loss) [48]. Their assessment is rather subjective and dependent on the type of T1-weighted sequence used and the MRI field strength [49]. These facts make longitudinal follow up of black holes rather difficult.

When detected, they seem to be correlated with disability progression [50] (studies showed that DMTs reduce the number of GdE lesions converting to permanent black holes, further proving their efficacy in preventing disability accumulation) [51,52,53,54]; their presence in CIS patients is associated with a higher risk of converting to RRMS [55]; and decreasing T1 values in black holes is associated with some degree of clinical improvement [56].

However, using black holes as prognosis biomarkers is still controversial. They weakly correlate with clinical severity on baseline evaluation, and studies investigating their predicting power have seen mixed results, ranging from no predicting power [57] to strongly correlating with EDSS worsening over 10 years (when assessing a combination of baseline black hole lesion count and increasing black hole lesion volume) [58]. Most positive studies on the matter did not retain their statistical significance on multivariate analysis when such an analysis was performed.

#### 3.5.2. T2/FLAIR Hyperintensities (White Matter Lesions—WMLs)

As stated by the ECTRIMS Focused Workshop on Aggressive MS, MRI T2 risk factors include high WML burden and infra-tentorial lesions.

The total lesion number is an important predictive biomarker, as a high WML burden is associated with disability progression, an aggressive course, conversion from CIS to RRMS [59], even predicting disability after 20 years [49,50], with more than 20 T2 lesion on baseline scan being considered a poor prognosis factor [13]. Baseline lesion count has also been shown to be correlated with EDSS and changes in lesion count have been shown to be correlated with changes in the EDSS [60,61,62,63].

The rate of lesion volume growth is three times higher in those who develop secondary progressive multiple sclerosis (SPMS) than in those who remain in relapsing–remitting multiple sclerosis (RRMS).

It is worth mentioning, however, that these correlations are frequently moderate at best [64,65].

When adding data from follow-up MRIs, the predictive power gets better. New lesions at 1 year and 3 years (particularly either GDE, spinal, or infra-tentorial lesions) are associated with an increased risk of developing SPMS after 15 years [66].

The predictive value of WML might be higher in the early phase of RRMS than later in the disease evolution. There is controversy of which feature of the WML holds more predictive power. Some authors consider their total number might be a better predictor than their location and degree of activity [67,68,69], while others consider the exact opposite to be true.

Regarding the localization of the lesions, infra-tentorial WMLs (symptomatic and asymptomatic) [59,70,71] and spinal lesions [72,73,74] have been demonstrated to predict the accumulation of disability in the short and medium term, following baseline MRI assessment in CIS and RRMS. For PPMS, lesions localized around motor tracts were the best predictor of disability [75].

WML total volume is associated with disability and motor and cognitive outcomes at long-term follow up. Shrinking WM lesions seem to hold no clinical relevance and are most likely due to the natural evolution of the lesions captured at different moments in time by the MRI scans [76].

While many automated WML-detecting software are available today—most accurate enough to be used in MS clinical studies and daily practice—an expert review is still advised for most scans [42,77]. New WM lesions are markers of poor disease control and can be used as biomarkers of poor prognosis starting with the baseline MRI (GdE vs. non-GdE lesions, see below).

One particular problem, however, lies with identifying and quantifying new lesions for patients with large, confluent lesions. A confluent lesion might hide two lesions connected by a single edge or dozens of connected lesions occupying large areas of white matter; in this scenario, a new lesion joining this confluence might be easily missed [78]. Longitudinal follow-up MRIs, even when properly administered, might be too far apart in time if performed yearly or less often. Studies have shown that even monthly or bi-monthly scans can reveal multiple new lesions that overlap in space. Automated software analysis might hold the key in such cases [79,80].

#### 3.5.3. Gadolinium-Enhancing (GdE) Lesions or Contrast-Enhancing Lesions (CELs)

A baseline MRI scan demonstrating ≥2 gadolinium-enhancing (GdE) lesions predicts the evolution to secondary progressive multiple sclerosis (SPMS) in 15 years, and the odds are increased if a new GDE lesion is found on the one-year follow-up MRI. They also hold predictive power regarding clinical disability, being positively correlated with EDSS at 15 years [66].

GdE lesions have also been found to be associated with the risk of future relapses and directly correlated with the relapse rate [55]. When two or more GDE lesions are present on the baseline MRI scan, they predict the risk of aggressive MS with a sensitivity of 0.73 and a specificity of 0.79 [11,13,81].

#### 3.5.4. Newer Concepts in WM Pathology

The classic MRI metrics regarding WML (described above) have proved only a moderate correlation to clinical activity and disability so far [82,83,84]. Pathology studies suggested that white matter is affected diffusely and not solely in the focal points we define as WML, but conventional MRI sequences were not able to fully capture this process.

Two new MRI concepts emerged—normal-appearing white matter (NAWM) and diffusely abnormal white matter (DAWM).

The concept of normal-appearing white matter (NAWM) refers to the normal looking, yet pathologically modified tissue around the white matter hyperintensities (WMH) on conventional MR images [85,86]. Altered metrics on advanced imaging sequences might hold precious information regarding prognosis, as it reflects inflammatory reactions that occur typically behind a grossly intact brain–blood barrier (BBB), possibly providing the missing link in correlating MRI WM lesions to neurological deficit [85].

When pathological changes are proven in NAWM on MRI studies of MS patients, (see below how that can be performed in vivo) they are associated with disability and act as an independent predictor of disability progression over 8 years [87,88].

The diffusely abnormal white matter (DAWM) has poorly defined boundaries and a signal intensity that lies between NAWM and classic WML. This MRI finding is very common when searched for, being present in a significant proportion (around 40%) of MS patients across all clinical subtypes [89].

Newer MRI techniques that have been deployed in MS research include magnetization transfer MRI (MT-MRI), diffusion tensor MRI (DT-MRI), and proton MR spectroscopy (^1^H-MRS).

MT-MRI can detect subtle brain tissue changes and is used to calculate an index of tissue integrity called the MT ratio. The reduction of this index in MS lesions and NAWM has been related to the percentage of residual axons and the degree of demyelination [90,91,92], and it may hold predictive power regarding the accumulation of clinical disability [93] (it may be worth mentioning that gray matter MT-MRI metrics have greater proven predictive power for disability over 8 years follow up than WM analysis) [87]. In fact, clinical studies have been using MT-MRI for assessing the efficacy of DMTs for years now [93,94,95].

Unfortunately, MT-MRI poses some technical challenges in inter-subject and inter-scanner variations, and guidelines regarding acquisition protocols have been elaborated since 2003 [96], with further error-correction significantly increasing the comparability of the obtained MTR values [97]. That being said, little has been published on this topic in the past five years, as MT-MRI sequences seem to have fallen out of focus.

DT-MRI sequences have been used in evaluating neuroaxonal integrity (including specific WM tracts). DTI-derived measures correlate with physical disability and cognitive impairment [98], and regarding predictive power, small studies showed that altered DT metrics in the NAWM of the corpus callosum may correlate with disability progression over 4 years in RRMS [99]. The same as with more conventional sequences, altered metrics of the WM hold some predictive power for disability progression and severity (and also future risk of MS) [100], while altered cortical metrics predict cognitive decline [101]. Pathological short-term DTI metrics of the thalamus have also been proven to predict the long-term accumulation of disability in PPMS [102]. Most studies agree that inter-subject and inter-scanner variability are within an acceptable range in the usual clinical settings [103,104,105].

Proton MR spectroscopy (^1^H-MRS) provides metabolic information regarding tissues in vivo. In MS, it has been used to analyze the chemical–pathological signature of lesions and NAWM. This sequence has brought to light that axonal damage is an early event in MS, occurring before the formation of T2/FLAIR-visible lesions, and that axonal loss is a major driver of disability in MS [106]. Due to its unparalleled capacity to estimate the concentrations of small, selected molecules in living tissues, many authors believe ^1^H-MRS could be used in detecting in vivo predictive biomarkers for MS patients [107]. Spectroscopic analysis of NAWM has consistent predictive power on brain atrophy and progression of disability (and perhaps even in predicting new WMLs) [108]. Unfortunately, technical difficulties have plagued the widespread adoption of proton MR spectroscopy, despite guidelines regarding acquisition protocols being elaborated since 2007 [109].

#### 3.5.5. Smoldering Lesions or Slowly Expanding Lesions (SELs)

Smoldering lesions, or chronic active lesions, or smoldering plaques, or slowly expanding lesions (SEL)—they go by many names—are WMLs that retain long-term chronic inflammatory activity at their edges [110,111,112].

Once visible only in pathological studies, in vivo detection has been proven possible using MRI scans. SELs can be divided in two categories: those that have a paramagnetic rim and those that do not.

Lesions that appear on MRI scans as having a paramagnetic rim around a non-GdE lesion (as seen on susceptibility-weighted imaging—SWI) are also called phase-rim lesions (PRLs). The first in vivo description of PRLs was specified on 7T MRI scans, but luckily recent articles proved that reliable detection can be achieved on both 3T and 1.5T scans as well (on the condition that a 3D acquisition is performed) [113,114]. We should say at this point that there is no clear consensus or guidelines regarding how we define a paramagnetic rim [115].

The second type of SELs, those that do not have a paramagnetic rim on the outer edge, represent around 60% of all smoldering lesions [116]. They can be detected in vivo using software analysis capable of proving expansion within existing T2-lesions on longitudinal MRI scans. Whether these two types of lesions are truly pathologically different or just different stages of chronically active lesions remains to be seen [11,38]. Other imaging modalities have also been tried for detecting these lesions, such as positron emission tomography (PET) using radiotracers specific to microglia/macrophages and sodium (23 Na) MRI, but their clinical use is so far very restricted [117].

The clinical significance of SELs is still controversial.

Some authors suggest that they might be more commonly found in (and a hallmark of) progressive MS (especially in patients with a disease duration of over 20 years who are also older than 50 years) [118]. Smoldering lesions are also considered an MRI risk factor for aggressive MS [111,119].

However, a recent meta-analysis showed that SELs are found in nearly half of all MS patients, and there is too little data available to decide whether they are more prevalent in (and a biomarker of) progressive forms of MS or not [120]. Many authors consider that the detection of SELs usually signals the risk of disability progression for both progressive and relapsing forms of the disease [121,122,123,124].

Some authors contest the predicting role of SELs, claiming there is no correlation between them and more aggressive types of MS [125,126,127]. The article by Arnold et al. [128] may be of particular interest on this topic.

#### 3.5.6. White Matter and Total Brain Volumetrics

Total brain volume measurements are tricky. MS patients, regardless of the clinical subtype, will present smaller brain volumes compared to healthy controls. Most of our knowledge comes from large cohorts of patients, where measures comparing group averages are usually reliable and have good intra- and inter-rater agreements, regardless of the software used. Unsurprisingly, higher magnetic field strengths produce better and more reliable measurements [129,130,131].

The problem with measuring total brain volume lies within individual assessments, for both baseline and follow-up scans, thus affecting our capacity to estimate the total brain atrophy on longitudinal scans for a certain, single, patient.

To the best of our knowledge, the average atrophy rate in MS patients is approximately 0.5–1.3% of the total brain volume per year, compared to 0.1–0.4% in healthy individuals. That means that the measurement error of brain atrophy needs to be very low in order to reliably detect changes over short periods of time [132,133].

One would correctly assume that different MRI machines (or different scanning protocols) will lead to significant variations between scans and final volumes. Unfortunately, even two identical machines using the same protocol can provide results that are significantly different regarding regional brain volumes [134]. Manufacturer, field strengths, pulse sequence, coil, data processing, filters, and patient positioning protocol are just some of the parameters that play a role in the final volumetrics provided [135,136,137].

In a study from 2010 [138], the authors scanned the same patient three times on six different scanners. The results showed an average combined variability of measurements of 4.80% (0.87–15.1%), with the conclusion that, for total brain volume, a cutoff for significant volume changes between two measurements in the same subject amounted to 1.4% on the same scanner and to 10.5% on different scanners (for an average atrophy rate of 0.5% in an MS patient, it would take at least 3 years on the same scanner to reach that cutoff, and 21 years if the follow up was performed on different scanners).

Demyelination and inflammation also play a role in brain volume beside neuronal and axonal loss, and we have to remember that there are many confounding factors such as volume reduction due to steroid treatment (such as that administered for a recent relapse) or pseudoatrophy, a decrease in brain volume during the first 6–12 months after starting a DMT. A phenomenon known as brain volume increase that occurs in a great proportion of MRI scans over short-term follow up (and is not associated with disease evolution) can also further complicate assessment [139].

We are still finding out what can impact total brain volume (and how). Drugs such as antipsychotics can lead to total brain volume loss; paroxetine and lithium may lead to the enlargement of DGM structures, as can other conditions (such as sleep apnea) and physiologic variables (time of day, menstrual cycle), which may impact the MRI volumetrics—aspects rarely taken into account in clinical studies of MS patients [140,141,142,143,144,145]. Contradictory data are reported on the effect of hydration for total brain volume measurements, with some authors claiming a difference of up to 0.36% between hydration states [146], while other authors state that it has no measurable effect [147].

Based on average group values rather than individual scans, volumetrics and all their derived measurements seem to hold great promise as prognosis biomarkers. A recent meta-analysis [148] showed that the effect of DMTs on disability progression was correlated to effects on both brain atrophy and active WMLs [149]. The effect on brain atrophy in RRMS was correlated to disability progression [150].

The 2018 ECTRIMS Focused Workshop on Aggressive MS (see above) included early discernible atrophy as an MRI predictive biomarker for aggressive MS based on a single study [151]. Bear in mind that it refers almost exclusively to deep gray matter changes in volume or cortical atrophy and not to whole brain volume or WM changes. In fact, this study found no predictive power for total brain volumes, and the correlations between whole brain atrophy and clinical changes were weak or absent.

This is a common occurrence. White matter volume (and atrophy) probably holds little predictive power, and most studies investigating this topic reached negative results [152,153]. This may be due to a number of reasons, some related to the pathological particularities of MS and at least to some degree due to all the technical limitations mentioned above.

At the moment of writing this article, the estimation of total brain atrophy in MS patients is probably possible only after several years of longitudinal follow up [154]. It would seem cautious to say that makes it very difficult to use brain atrophy as a prognosis biomarker at individual level, especially in the early stages of the disease, when it would be most needed [76,155,156,157]. The same cannot be said regarding GM volumetrics.

As a conclusion, long-term outcomes (with an emphasis on “long”) in MS patients correlate moderately with WM MRI metrics, suggesting that different mechanisms might be at play in the natural course of MS [158].

### 3.6. Spinal Atrophy

The spinal cord (particularly, the cervical segment) is more atrophied in MS patients versus that in healthy controls, with a greater atrophy rate than the total brain one, and greater in PPMS rather than RRMS [159].

Reliable longitudinal measurements are possible using the standardized cross-sectional area of the upper cervical cord [160]. Automated MRI measurements including total volume and individual white and gray matter volumes are also possible today [161].

Evidence is sparse regarding clinical outcomes and particularly regarding prognosis implications for individual-level longitudinal follow up, but recent data show that even a small increase in the spinal atrophy rate is associated with a significantly increased risk of disability progression [162].

### 3.7. Evaluating Gray Matter (GM) Pathology in MS

We have known for decades that postmortem cerebral histological examinations of MS patients reveal cortical demyelination (and pathology) that is often more extensive than white matter demyelination [163,164,165].

Research has shown that GM abnormalities seem to occur from the first clinical demyelinating event (clinically isolated syndrome—CIS), and their presence predicts the conversion to MS, as well as the progressive accrual of disability. GM pathology might be the earliest manifestation of MS, and we have proof that GM atrophy is more severe than WM atrophy early in the course of the disease [152].

GM pathology also elegantly explains the observed dissociation between markers of inflammatory demyelination (relapses, WML gadolinium enhancement, and WML burden) and disease progression [165]. Physical disability, fatigue, and cognitive impairment in MS all seem to be tied to GM pathology as well [166,167,168].

This accumulating knowledge generated a shift toward considering MS as a pathology involving both WM and GM (the 2017 McDonalds criteria included, for the first time, cortical lesions as proof of dissemination in space) [169].

GM pathology can currently be evaluated in two ways on MRI scans—GM lesions and GM volumetrics. Most studies investigate deep gray matter (DGM) and cortical GM separately.

#### 3.7.1. Deep Gray Matter (DGM) Pathology

Most authors agree that counting lesions of the DGM is difficult, while also not showing a significant correlation with MS severity (at least not on 3T scans) [170]. Some studies found that the fastest regional decline in tissue volume over time was observed in the DGM in all clinical phenotypes of MS. What is more, in a large cohort of patients, only the rate of volume loss in the DGM was associated with disability accumulation and not WM or cortical loss. The DGM atrophy progression was not different between any DMTs or for not being under treatment, and it predicted future EDSS progression.

DGM atrophy has also been shown to be associated with developing definite MS for CIS patients, to predict disability progression in early RRMS and disability progression in PPMS [102,171,172,173]. In a recent study with a limited number of patients, isolated thalamic atrophy predicted a higher risk for not reaching 2-year NEDA-3 and for EDSS increase [174].

Most of the cited studies relied on group-level brain volume estimates (including DGM) for statistical power. As is the situation in this study [151] as well, in AUC analysis, at the individual level, DGM volume lacked prognostic value due to the high variability of these metrics (typical of volumetric MRI studies). Scans from 1.5T MRIs are also significantly less reliable [175]. It is for these reasons that some authors felt that we are not ready yet for the use of DGM volumetrics at the individual level at this point in time [176]. Recently, however, single time point, individual assessment of DGM atrophy has been proven possible [174,177], opening the way to a new subtype of MRI predictive biomarkers.

#### 3.7.2. Cortical Lesions

Demonstrating cortical lesions in vivo is quite difficult. Only about 30% to 50% of histopathologically confirmed lesions can be detected at 7T MRI imaging, while the industry standard remains at 1.5 Tesla for most of the world [178]. In a postmortem tissue–MRI correlation study, the 1.5T MRI T2-weighted images captured around 3% of cortical lesions and FLAIR captured roughly 5% [179]. Double inversion recovery (DIR) sequences greatly improved cortical lesion detection but still missed 82% of the histopathologically confirmed ones for 1.5T scans and marginally better for 3T [180,181].

In an optimistic take on the “glass half-full”, we can still use cortical lesions as an MS biomarker, if we focus on the lesions that we can see. In one study [182], for example, the authors found that after a mean follow up duration of 1.5 years, more patients developed cortical lesions rather than WMLs. Cortical lesion accrual was greater in patients with secondary progressive multiple sclerosis (SPMS) than in those with relapse-remitting multiple sclerosis (RRMS), and total cortical lesion volume independently predicted baseline EDSS and EDSS changes at follow up (data coming from 7T MRI scans but still of interest).

Cortical lesions are also correlated to disability and cognitive impairment in early MS stages and show greater predictive power regarding clinical outcome and disability progression than WMLs [183,184,185].

#### 3.7.3. Cortical Atrophy

While in vivo cortical lesion detection might still be suboptimal, we are arguably better at measuring GM atrophy. This was initially (think late 1990s) performed manually, by trained readers, using gross measures such as whole brain parenchymal volume—a very time-consuming task, unfit for studying large cohorts of patients [179]. Automated software that could perform this task was later developed, leading to a much more refined analysis of the GM volumetrics (segmentation, deep GM/cortical individual volumes, cortical thickness, regional cortical volumes, etc.). This software proved to be reproducible, with satisfactory intra- and inter-rater, and even inter-center, agreement (and better than those for total brain and WM volumetrics) [54,186,187,188,189,190,191,192].

We know today that GM atrophy is associated with disease progression and is markedly worse in patients converting to SPMS from RRMS [168,193,194]. Gray matter (and whole brain volume) atrophy seems to predict an increased risk of developing RRMS following a clinically isolated syndrome (CIS) [195].

GM atrophy (and not WM atrophy) might be responsible for total brain volume loss despite clinical and disability apparent stability under DMTs [196].

The clinical impact of volume ratio between GM and WM (namely normal-aspect white matter—NAWM) was also investigated. Though retrospective, a study [197] analyzed a cohort of 149 patients newly diagnosed with RRMS who had been followed up for 10 years. The ratio between GM and WM total volume was corelated with EDSS progression, and individuals who had higher GM:NAWM ratio at diagnosis had a 90% lower rate of reaching EDSS 4.0 and of converting to SPMS. As a reminder, most MRIs were performed at 1.5T, with only a fraction of the patients undergoing 3T MRI scans.

Cortical atrophy is also considered to be an MRI predictive biomarker for aggressive MS [174].

### 3.8. A Brief Glance at Prognosis Scores

Before finishing this review, it is perhaps worth briefly mentioning prediction scores in MS.

What if it were not one single element that we should use as a prediction tool, but rather multiple factors that are known to be associated with a poor outcome? This concept, of creating prognosis scores in MS, has been around for a long, long time.

Many authors tried to use data derived from large cohorts of patients (some of whom had a natural history of the disease) and create a prediction model for long-term prognosis, based mainly on clinical and MRI data available in the early stages of diagnosis (usually from baseline to one-year follow up).

A systemized review of prognosis scores that had been published up to August 2019 (with over 30 scores included in the analysis) concluded that “Although a number of prediction models for RRMS have been reported, most are at high risk of bias and lack external validation and impact analysis, restricting their application to routine clinical practice” [198].

Overall, the most robust predictors of poor prognosis across these scores seem to be early sphincter involvement, higher baseline disability, and certain MRI measurements (brain atrophy rate and T2 lesions number and volume). Unfortunately, these rely on established damage and, therefore, are not ideal prognostic markers of the future [198].

Published in May 2021, the Secondary Progressive Risk Score (SP-RiSc) by Calabrese et al. [199] was not included in the review mentioned earlier. This score is different from its predecessors as it heavily relies on cortical pathology, which greatly enhances its predictive accuracy. The predictors included are age, baseline EDSS, cortical lesions number at baseline and 2-year follow up (obviously, the ones we see), WM lesions number, cerebellar cortical volume at baseline and 2-year follow up, global cortical thickness at baseline and 2-year follow up).

What is perhaps most important is that SP-RiSC performs with great accuracy, sensibility, and sensitivity at the individual level, with scores of ≥17.7 indicating a 92% probability of converting to SPMS within 10 years from the disease diagnosis. In contrast, patients with SP-RiSc < 17.7 had an 87% probability of remaining in the relapsing–remitting phase.

Many other scores have been developed for MS, including scores predicting DMT response, but they are not of interest for the sole purpose of this review [200,201].

## 4. Discussion

MRI scans have transformed our understanding and approach to MS. After almost four decades of use, MRI techniques are still evolving, and we are still learning new ways to implement classic and novel metrics in the diagnostic process, in guiding treatment, and in offering a prognosis.

Due to its prevalence and impact on the active population, MS has been for decades one of the most dynamic scenes in modern medicine. The large volume of high-quality research has allowed us to keep improving our capacity to corelate MRI scan results to clinical evolution and pathological studies and derive from these data much needed prognosis biomarkers.

While much promising data have been published in recent years, we must take this predictive capacity with caution, as almost all metrics in use today have their own pitfalls and shortcomings.

The advent of complex automated analysis has opened new horizons in MRI metrics. Volumetrics; segmental atrophy longitudinal follow up; and hybrid techniques for identifying smoldering lesions, quantifying DAWM, and the detection of cortical lesions are just some of the recent findings that changed the way we think of MS. With ever improving software detection capacity, the future holds much promise in regard to deriving even more data from this paramount investigation.

## 5. Conclusions

Our critical review brings under scrutiny classic and novel MRI metrics in use today as predictive biomarkers for multiple sclerosis. While some of these parameters have passed the test of time and seem to be associated with a reliable predictive power, some are still better interpreted with caution. We hope this will serve as a reminder of how vast a resource we have on our hands in this versatile tool—it is up to us to make use of it.

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
