# Peer review of "Predictive MRI Biomarkers in MS—A Critical Review"

_medicina, 2022, doi:10.3390/medicina58030377_

Round 1

Reviewer 1 Report

This is an important review with regard to MS prognostic biomarkers. The authors did a nice job reviewing the literature and organizing their findings into this manuscript. However, there are some minor issues like typos and improper word use. Please read below for some suggestions to revise the current version: 

Abstract:

Change: "the role of MRI measurements as a predictive biomarker in multiple sclerosis (MS) patients"

To "the potential use of MRI measurements as prognostic biomarkers in multiple sclerosis (MS) patients"

This sentence seems off/unclear: "for both conventional measurements, as well as more novel techniques"

Introduction:

The following selectance needs to be rewritten: "but our best predictive biomarker in MS to date might reside in the Magnetic Resonance Imaging (MRI) measurements. "

L.27: PRGONSIS value --> PROGNOSTIC value

Materials and Methods:

The authors could describe their methods better by detailing the exact searches they conducted and the exact terms they used when conducting each search on each database.

Results:

L.33: The need for PROGNOSIS biomarkers in MS  --> PROGNOSTIC

Rewrite: "And while some cases might be clear regarding where they classify on this scale of disease activity, some will not, and predicting which way a patient’s evolution will go following diagnosis, is a very difficult task. "

I recommend using  disease progression instead of patient's evolution.

Unclear, rewrite: So active it becomes considered “aggressive”? We don’t know. 

L.127: change from "stating that patients THAT had" to "stating that patients WHO had"

L.134: DIAGNOSTIC process into DIAGNOSTIC process 

L.158: change "We mustn’t " to “we must not”

  1. 483: Add “A” to the title of this section “A Brief …”

Conclusions:

L.533: change “and the “spotlightS” to “and the spotlight”

 Recent data SEEMS to suggest it would be better if we could start most patients on 37  -- SEEM

Author Response

DIAGNOSTIC process

Change: "the role of MRI measurements as a predictive biomarker in multiple sclerosis (MS) patients"

Was changed as suggested to "the potential use of MRI measurements as prognostic biomarkers in multiple sclerosis (MS) patients"

This sentence seems off/unclear: "for both conventional measurements, as well as more novel techniques" - added a brief list of novel techniques at the end of the sentence

Introduction:

The following selectance needs to be rewritten: "but our best predictive biomarker in MS to date might reside in the Magnetic Resonance Imaging (MRI) measurements. " - rewritten for clarity

L.27: PRGONSIS value --> PROGNOSTIC value - corrected

Materials and Methods:

The authors could describe their methods better by detailing the exact searches they conducted and the exact terms they used when conducting each search on each database. - this section was completely rewritten in accordance to indications

Results:

L.33: The need for PROGNOSIS biomarkers in MS  --> PROGNOSTIC - corrected

Rewrite: "And while some cases might be clear regarding where they classify on this scale of disease activity, some will not, and predicting which way a patient’s evolution will go following diagnosis, is a very difficult task. " - this paragraph was completely rewritten

Unclear, rewrite: So active it becomes considered “aggressive”? We don’t know. - reworded so that it only addresses uncertainties regarding aggressive MS

L.127: change from "stating that patients THAT had" to "stating that patients WHO had" - changed accordingly

L.134: DIAGNOSTIC process into DIAGNOSTIC process  - changed accordingly

L.158: change "We mustn’t " to “we must not” - changed accordingly

483: Add “A” to the title of this section “A Brief …” - changed accordingly

Conclusions:

L.533: change “and the “spotlightS” to “and the spotlight”  - changed accordingly

 Recent data SEEMS to suggest it would be better if we could start most patients on 37  -- SEEMDIAGNOSTIC process  - changed accordingly

Reviewer 2 Report

The article is informative and well written.

I would suggest to improve the introduction, in order to describe clinical and non-clinical aspect of MS. Also I would suggest to add in introduction a part on MRI including advances to help diagnosis of MS. Even if the article is more focused on MRI, I think it's interesting for the reader to have a brief description of MS and a clear view of MRI as diagnosis tool. 

In conclusion or Discussion section I would discuss a perpective of such technology in the future, what would it be interesting to improve, which parameters to include to have an accurate diagnosis.

Author Response

I would suggest to improve the introduction, in order to describe clinical and non-clinical aspect of MS. Also I would suggest to add in introduction a part on MRI including advances to help diagnosis of MS. Even if the article is more focused on MRI, I think it's interesting for the reader to have a brief description of MS and a clear view of MRI as diagnosis tool.

We added some data in the introduction. We feel that the introduction is already rather lengthy, and the main focus of the review is on MRI predictive biomarkers, not on diagnosis. Tempting as it may be to expand into other areas of MS pathology, we would rather not add this section on MRI as a diagnosis tool, if that is ok.

In conclusion or Discussion section I would discuss a perspective of such technology in the future, what would it be interesting to improve, which parameters to include to have an accurate diagnosis.

We have added the following to Discussions:

The advent of complex automated analysis has opened new horizons in MRI metrics. Volumetrics, segmental atrophy longitudinal follow-up, hybrid techniques for identifying smoldering lesions, quantifying DAWM, detection of cortical lesions are just some of the recent findings that changed the way we think of MS. With ever improving software detection capacity, the future holds much promise in regards to de-riving even more data from this paramount investigation.